evolution, genetics, cognition

language diversity, auditory processing, genetic variation

**Author for correspondence:**
Jeffrey R. Gruen
e-mail: jeffrey.gruen@yale.edu

†Both authors contributed equally.

# *DCDC2* READ1 regulatory element: how temporal processing differences may shape language

Kevin Tang[1,†], Mellissa M. C. DeMille[2,†], Jan C. Frijters[4] and Jeffrey R. Gruen[2,3]

[1]Department of Linguistics, University of Florida, Gainesville, FL 32611-5454, USA
[2]Department of Pediatrics, and [3]Department of Genetics, Yale University School of Medicine, New Haven, CT 06520, USA
[4]Child and Youth Studies, Brock University, St. Catherine's, Ontario, Canada L2S 3A1

  KT, 0000-0001-7382-9344; MMCD, 0000-0001-7068-395X; JCF, 0000-0001-5625-4892; JRG, 0000-0001-7640-2071

Classic linguistic theory ascribes language change and diversity to population migrations, conquests, and geographical isolation, with the assumption that human populations have equivalent language processing abilities. We hypothesize that spectral and temporal characteristics make some consonant manners vulnerable to differences in temporal precision associated with specific population allele frequencies. To test this hypothesis, we modelled association between RU1-1 alleles of *DCDC2* and manner of articulation in 51 populations spanning five continents, and adjusting for geographical proximity, and genetic and linguistic relatedness. RU1-1 alleles, acting through increased expression of *DCDC2*, appear to increase auditory processing precision that enhances stop-consonant discrimination, favouring retention in some populations and loss by others. These findings enhance classical linguistic theories by adding a genetic dimension, which until recently, has not been considered to be a significant catalyst for language change.

## 1. Introduction

Historical events and population interactions have shaped the differences among the over 7000 languages spoken in the world today. Genetics strongly influences functions essential for oral communication, articulation [1], hearing, [2] and phonological processing [3]. Phonological processing is the ability to identify meaningful information in a stream of speech sounds, which requires faithful spectral and temporal encoding in the auditory cortex [4]. There are several genes that modify this encoding, including *DCDC2*, which we previously showed has population effects on phoneme inventories [5].

   *DCDC2* is associated with reading and language performance and related disorders of dyslexia, specific language impairment, and speech sound disorders [6]. READ1, a regulatory element encoded in the second intron of *DCDC2*, is also associated with normal variation in phonological processing [7]. It has over 40 alleles that differentially enhance transcription through the *DCDC2* promoter [8], which can be divided into three groups based on the presence (RU1-2) or absence (RU1-1) of an ancient 13 base duplication, or a 2.4 kb microdeletion encompassing the entire READ1 sequence. We previously showed that the frequency of RU1-1 alleles correlates with the number of consonants in the languages of 43 populations from five continents and major language families [7]. Functionally, in rodent models *Dcdc2* modifies speech-sound discrimination between consonants [9,10] and is critical for temporal precision of stimulus-driven action potential firing and baseline excitability in neurons of the auditory cortex [11,12]. These studies suggest that through their effect on *DCDC2* transcription, some READ1 alleles can enhance temporal precision and speech sound discrimination to favour retention or acquisition of selected consonantal contrasts. In the present study, we build

**Table 1.** Manner of articulation examples.

| manner of articulation | | frequency energy | English phonemes | English examples |
|---|---|---|---|---|
| obstruents | stop | high | /p/, /t/, /k/ | pea, tea, key |
| | fricative | | /f/, /s/, /h/, | fee, see, he |
| | affricate | | /t͡ʃ/, /d͡ʒ/ | cheese, judge |
| sonorants | nasal | low | /m/, /n/, /ŋ/ | meat, neat, thing |
| | approximant | | /j/, /w/, /r/, /l/ | yes, way, read, lead |

on and expand our previous results to investigate the relationship between the population frequency of RU1-1 and the numbers of specific manners of consonants used in a language. Consonant manners differ in their temporal and spectral characteristics. The perception of these different manners may be more or less sensitive to the differences in temporal processing conferred by RU1-1 specific regulation of *DCDC2*, favouring retention or acquisition of selected consonantal contrasts.

Consonants are described by three cardinal articulatory features: manner of articulation, place of articulation, and voicing [13]. Manner of articulation is the configuration and interaction of the tongue, lips, palate, and nose when making a speech sound and is subdivided into obstruents and sonorants (table 1). Obstruents and sonorants rely on high and low acoustic frequency energy respectively. Obstruents are further divided into three manners called stops, fricatives, and affricates. Stops and affricates require a complete closure of the airway; fricatives are produced by partial closure. The high frequency energy of fricatives and affricates are more robust to noise in phonological processing, whereas the transient release bursts of stop and affricates are less robust [14]. The two sonorant manners are nasals and approximants. All of these manners of articulation result in acoustic spectrograms that can be recorded from the auditory cortex as distinct neurograms [15].

We hypothesize that the acoustic differences among consonants create perceptual challenges that may make some consonants more vulnerable to loss of temporal precision than others [14] and thus more likely to be associated with RU1-1 alleles.

## 2. Methods and results

We modelled association between RU1-1 alleles and manner of articulation in 51 populations, spanning five continents and all major language families while accounting for geographical proximity, and genetic and linguistic relatedness. For the current analysis we added nine populations from the 1000 Genomes Project (1KG) [16] to the 43 populations we used in our previous study (electronic supplementary material, table S1). We then correlated the number of consonants and vowels used in each language against the frequencies of three groups of READ1 alleles: RU1-1, RU1-2, and the READ1 2.4 kb microdeletion. Using this expanded set of populations showed that the number of consonants, but not vowels, correlated with the frequency of RU1-1 ($r = 0.314$, $p = 0.025$) in their respective population, replicating our previous results.

To identify the specific linguistic features that contribute to this association, we then correlated the consonants that comprise the five manners of articulation with the frequency of RU1-1 (table 2) in all 51 populations. The number of stops had the

**Table 2.** Uncorrected correlation of manner of articulation with the frequency of RU1-1. Level of significance: ** ($p \leq 0.01$).

| manner | r | p-value |
|---|---|---|
| stops | 0.398 | 0.0038** |
| fricatives | 0.263 | 0.0627 |
| affricates | 0.0914 | 0.524 |
| nasals | −0.19 | 0.182 |
| approximants | 0.173 | 0.224 |

strongest association at $r = 0.398$ ($p = 0.0038$), however RU1-1 frequency and numbers of stops clustered by regional location (figure 1). We, therefore, controlled for the effects of genetic relatedness, geographical proximity, and linguistic relatedness between populations (see electronic supplementary material for details). For genetic relatedness, 164 informative single nucleotide polymorphisms (SNPs), independent from RU1-1, were used to compute pairwise $F_{st}$ values between populations. For geographical proximity, we modelled migratory distances between populations and the putative location of human origin using great circle distances along with migratory waypoints, which increase the accuracy of these distances. For linguistic relatedness, the sound inventory of the populations was used to compute the degree of inventory overlap between populations.

To better understand these three effects, we evaluated how genetic relatedness, geographical proximity, and linguistic relatedness correlate with each other using the Mantel test, a statistical test of the correlation between distance matrices. Statistical significance was determined using 1000 permutations. Genetic relatedness and geographical proximity are the most strongly correlated with $r = 0.563$ ($p = 0.001$), while linguistic relatedness does not correlate with either genetic relatedness ($r = −0.0771$, $p = 0.860$) or geographical proximity ($r = 0.01434$, $p = 0.448$). The result of the Mantel tests suggests that all three effects should be taken into account in the modelling of the association between the five manners of articulation and RU1-1, given that linguistic relatedness is relatively independent from geographical proximity and genetic relatedness, and geographical proximity and genetic relatedness are only moderately correlated.

To control for confounding due to effects of genetic relatedness, geographical proximity, and linguistic relatedness, we created a generalized linear regression model and included the major principal components (PCs) for each of the three sets of pairwise distances between populations. Insufficient populations were available to simultaneously examine all five manners, vowels, tones, and the control variables (the PCs) in the regression model [17]. Therefore, we performed

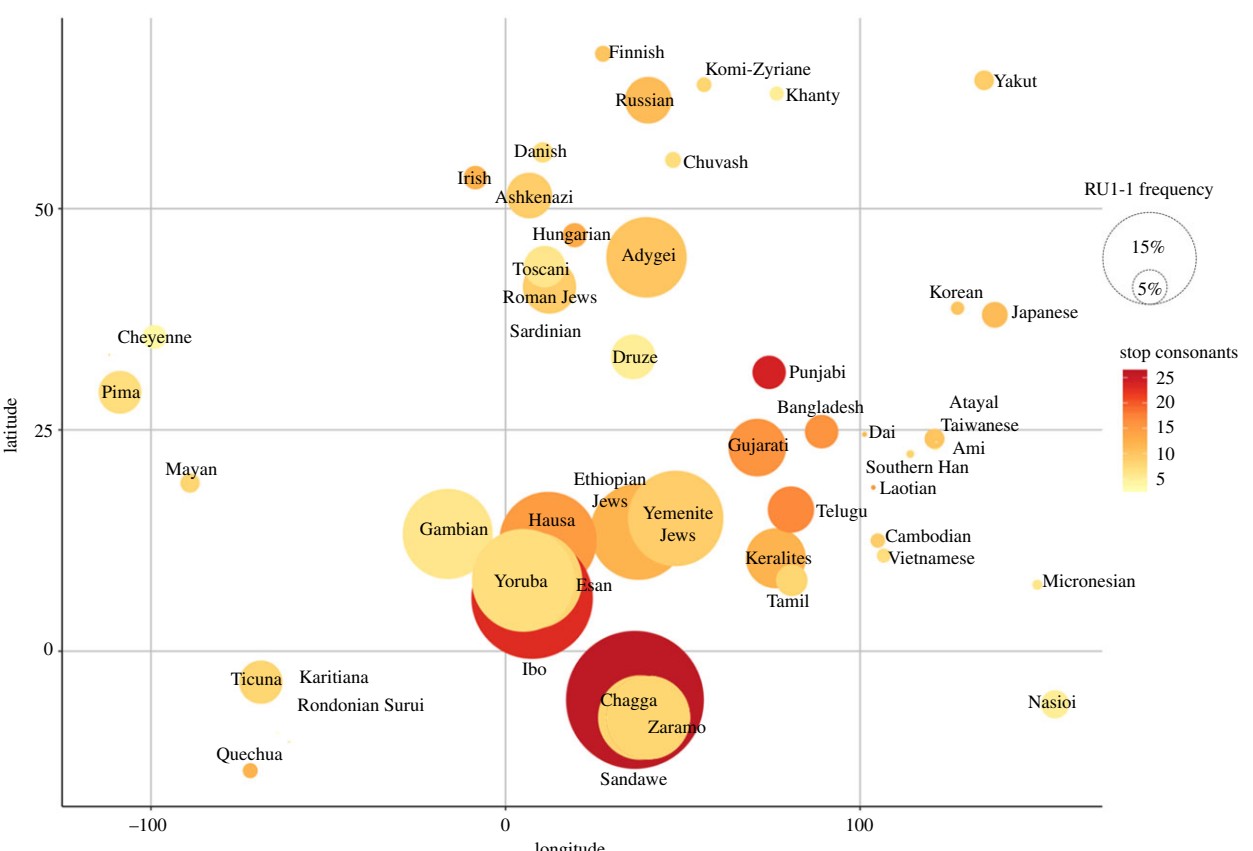

**Figure 1.** Fifty-one populations, plotted by longitude and latitude. The size of the circles is defined by the frequency of RU1-1 in that population. Circles are coloured by the number of stops. (Online version in colour.)

**Table 3.** Regression summary for stops and nasals. SE: standard error; *t*-value; $CI_{Lower}$ and $CI_{Upper}$: 95% confidence intervals of the coefficient from bootstrapping; *p*-value from 10 000 permutations. Level of significance: · (nominally significant, $p \leq 0.1$), * ($p \leq 0.05$).

|  | beta-value | s.e. | *t*-value | $CI_{Lower}$ | $CI_{Upper}$ | *p*-value |
|---|---|---|---|---|---|---|
| stops | 0.0113 | 0.0049 | 2.329 | 0.0021 | 0.0208 | 0.0318* |
| nasals | −0.0089 | 0.0053 | −1.674 | −0.0192 | 0.0026 | 0.0798· |

nested model comparisons with all possible combinations of the five manners and the number of vowels and tones to avoid over-fitting, and found that the most parsimonious model contained stops and nasals:

$$RU1\text{-}1 \sim \text{stops} + \text{nasals} + \text{genetic PC-1} + \text{genetic PC-2}$$
$$+ \text{genetic PC-3} + \text{geographical PC-1} + \text{geographical PC-2}$$
$$+ \text{geographical PC-3} + \text{linguistic PC-1} + \text{linguistic PC-2}$$
$$+ \text{linguistic PC-3} + \text{linguistic PC-4}$$

$$(2.1)$$

Statistical significance was estimated using 10 000 permutations. RU1-1 was significantly associated with stops and only nominally associated with nasals, but in opposite directions (table 3 and figure 2). To evaluate whether the associations were robust across different population subsets we performed a series of leave-*k*-out analyses by population from *k* = 1 (jackknife) to *k* = 30 (electronic supplementary material, tables S4). The directions of the beta-values remained positive for stops and negative for nasals for 95% of the subsets even when almost 50% (*k* = 26 for stops and *k* = 23 for nasals)

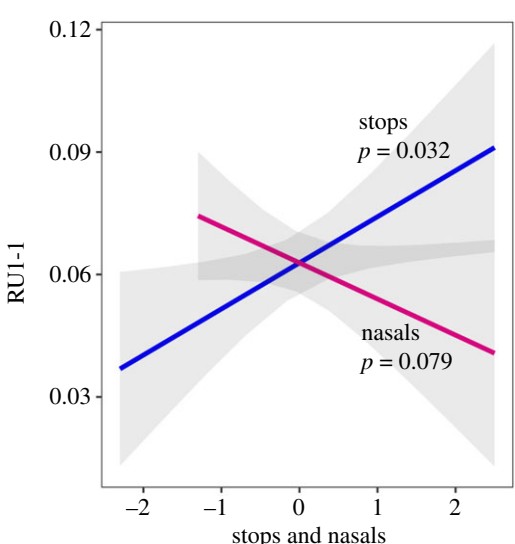

**Figure 2.** The relationship between RU1-1 frequency and the number of consonants by stops and nasals (log-transformed, *z*-scores) as fitted in the best regression model (equation (2.1), table 3). Shaded regions are 95% confidence intervals. (Online version in colour.)

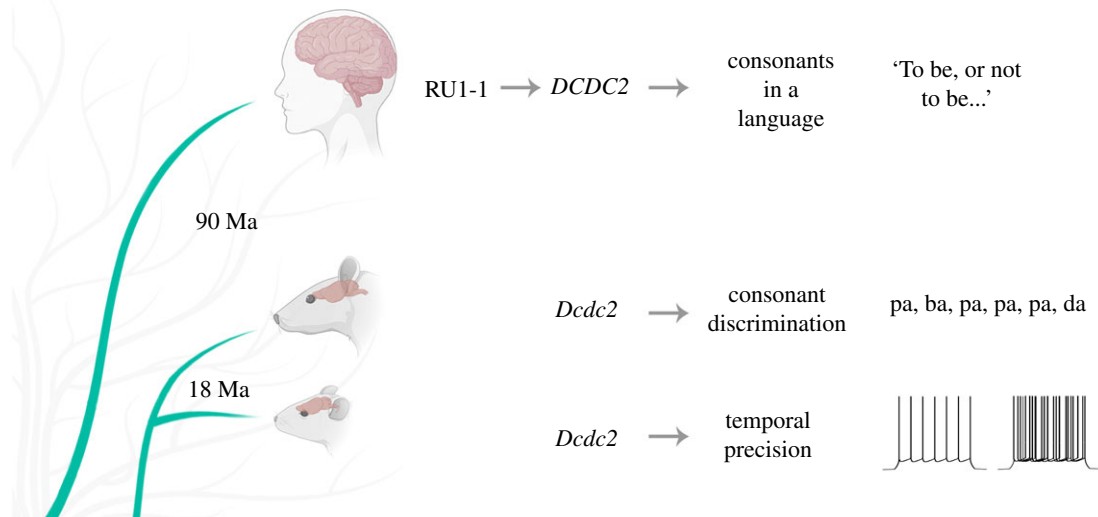

**Figure 3.** Experimental lines of evidence for the role of DCDC2/RU1-1 in the number of consonants in a language. Dcdc2 KD rat has reduced consonant discrimination in a stream of speech sounds. Dcdc2 KO mouse has reduced temporal precision. Created with biorender.com. (Online version in colour.)

were excluded, showing that the associations were not driven by particular subsets.

## 3. Discussion

Our results revealed that RU1-1 is associated positively with stops and negatively with nasals in 51 populations, adjusting for geographical proximity, and genetic and linguistic relatedness. To contextualize our results, we examined consonant processing in published animal models. Recordings from the auditory cortex of wild-type rats in response to different consonant manners showed that neurograms from stops have the sharpest onset peaks, are the most distinctive manner of articulation, and can be distinguished better than other manners in the first 40 milliseconds (ms) [9]. By contrast, while it takes longer to discriminate between pairs of nasals (greater than 40 ms), they are more robust to loss of millisecond temporal precision compared to stops, which are more sensitive.

Disruption of *Dcdc2* in rodents also supports a prominent role in differential perception of stops and nasals. RNAi knockdown (KD) of *Dcdc2* in rats leads to an increase in excitability in neurons located in the auditory cortex, as evidenced by both spontaneous and stimulus-driven action potentials and shorter onset latency in firing [18]. The correlate in the *Dcdc2* knockout (KO) mouse also shows an increase in neuronal excitability demonstrated by an elevated resting membrane potential in whole-cell patch clamp studies of brain slices [11]. KO mice also have decreased temporal precision of stimulus-driven action potential firing. This decrease is mediated by increased expression of *Grin2B*, a subunit of the N-methyl-D-aspartate receptor (NMDAR), and restored by an NMDAR antagonist. Behaviorally, *Dcdc2* KD rats have defective consonant detection in speech streams presented at all speeds, and altered neurograms at very high speeds recorded from the auditory cortex. *Dcdc2* KO mice have impaired rapid auditory processing [12]. These studies show that in rodent models, *Dcdc2* is important for potentiating baseline excitability and temporal precision in neurons of the rodent auditory cortex, which are critical for the sensitive and accurate perception of specific sound targets, and consonants in particular. Human *DCDC2* promoter

reporter-gene constructs show that the most frequent RU1-1 allele confers higher expression than the most frequent RU1-2 allele [8]. This suggests that RU1-1 alleles increase *DCDC2* expression, and as suggested by the rodent experiments, could lead to increased temporal precision in the auditory cortex and enhanced consonant discrimination (figure 3).

Together, these data suggest that RU1-1 alleles acting through increased expression of *DCDC2*, increase auditory processing precision that enhances stop-consonant discrimination. In this model, populations with higher RU1-1 allele prevalence would have enhanced ability to discriminate between similar stop-consonants, and over time, change the phoneme inventory of the language by addition or retention of words with similar stops, expand the stop-consonant repertoire, and increase the total number of stop-consonants relative to other sounds. When RU1-1 prevalence is low in a population, fewer individuals would have the ability to finely discriminate between stops. Thus, stop consonants would potentially be fewer, and nasals, which do not require enhanced temporal precision and are robust to noise, would be more likely to be recruited or retained. This would account for the inverse relationship of stops and nasals that we observed (figure 2).

Linguistic theory holds that listeners play an important role in shaping the sound structure of language [19,20] through perceptual biases that introduce errors [21], and result in vocal compensation to ensure effective communication [22,23]. For example, a high prevalence of chronic ear infections (up to 90%) in some Australian Aborigine populations is thought to be the cause of a phonemic inventory lacking sounds that depend on acoustic frequencies impacted by the infections [24,25]. Differences in the auditory processing of stops and nasals are more subtle than loss of acoustic frequencies, but nonetheless have physiologic correlates in evoked response potentials measured in the auditory cortex [15].

While the results of the current analyses show a significant association of stops and nasals with RU1-1 at the population level, it is important to note that subjects were not phenotyped individually for differences in speech perception. Additionally, language assignment has potential for errors and phonemic inventory counts can differ between studies and sub-populations. While our genetic samples were chosen

to be representative of the population as a whole, hidden population stratification not accounted for by the PC corrections could distort the findings. Although the overall effect size of RU1-1 on language change is likely to be small, subtle cognitive biases can be amplified through cultural transmission to shape the structure of languages over time; simulated models of evolution show that a small difference (as little as 5%) in population prevalence of a bias in favour of a linguistic change can disproportionately increase the number of speakers by up to 45% [26]. In addition, the link between *DCDC2* expression and stop-consonant discrimination relies on published experiments in rodents. Further studies in humans are necessary to demonstrate a more direct genotype–phenotype connection. It should also be noted that although we interpret the association between RU1-1 and stop-consonant description as a possible driver for phoneme inventory change among populations, a strict interpretation of the analysis means that the inverse relationship is also possible and that certain phoneme inventories may have favoured genetic selection. However, we view this as unlikely, given arguments from computational simulations of language change and experimental evidence from human artificial language learning and song birds [27].

## 4. Conclusion

Language is continuously evolving to meet the needs both of the speaker and of the listener [28]. The needs of the speaker include balancing ease of articulation and communicative success. The needs of the listener include ease of decoding by having a signal that is robust to noise. Stop consonants perform well when background noise is low and listeners with RU1-1 alleles have greater capacity to discriminate between them. When background noise is prominent, listeners without RU1-1 alleles may have reduced capacity to discriminate between stops—nasals are preferred because of their robustness to noise. The nature of these consonant manners and the direction of their relationships with RU1-1 supports an account of how languages evolve to maintain phonemes that are robust to auditory processing constraints [20]. These findings enhance classical linguistic theories on the evolution of language shaped by historical migrations [29,30], conquests [31], admixtures [32], geographical isolation [33–37], diet [38], and communication efficiencies [28,39] by adding a genetic dimension, which until recently, has not been considered to be a significant catalyst for language change.

Data accessibility. All data and our analyses are available in the electronic supplementary materials and are available via the Dryad Digital repository https://dx.doi.org/10.5061/dryad.hdr7sqvf4 [40].

Authors' contributions. Conceptualization, K.T., M.M.C.D., and J.R.G.; investigation, K.T., M.M.C.D., and J.C.F.; writing—original draft, K.T and M.M.C.D.; writing—editing, K.T., M.M.C.D., and J.R.G.; funding acquisition and supervision: J.R.G.

Competing interests. Yale University has applied for a patent covering READ1. J.R.G. is an inventor named on the patent application.

Funding. Support for J.R.G., M.M.C.D., and J.C.F. was provided by The Manton Foundation. Support for J.R.G. was provided by NIH/ Eunice Kennedy Shriver National Institute of Child Health and Human Development (NICHD) P50 HD027802.

Acknowledgements. We wish to thank Dongnhu Truong, Jeffrey Malins, and Andrew Adams for helpful discussions. We thank Bill Speed for providing a merged dataset of Kidd laboratory and 1KG SNPs and Julian DeMille for his assistance with data extraction.

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
