## [Reviewer comments · Proceedings of the Royal Society B: Biological Sciences]

Review History

RSPB-2019-2712.R0 (Original submission)

Review form: Reviewer 1

Recommendation

Accept with minor revision (please list in comments)

Scientific importance: Is the manuscript an original and important contribution to its field?

Excellent

General interest: Is the paper of sufficient general interest?

Excellent

Quality of the paper: Is the overall quality of the paper suitable?

Excellent

Is the length of the paper justified?

Yes

Should the paper be seen by a specialist statistical reviewer?

No

Do you have any concerns about statistical analyses in this paper? If so, please specify them explicitly in your report.

No

It is a condition of publication that authors make their supporting data, code and materials available - either as supplementary material or hosted in an external repository. Please rate, if applicable, the supporting data on the following criteria.

Is it accessible?

Yes

Is it clear?

Yes

Is it adequate?

Yes

Do you have any ethical concerns with this paper?

No

Comments to the Author

The article integrates data from Genetics and Linguistics to put forward an hypothesis about the effect of weak genetic biases in language diversity. In particular, the authors propose that variation in the READ1 regulatory element, in the DCDC2 gene, is correlated with stop-consonant discrimination. Small effects of genetics in auditory processing would then be amplified into language diversity by cultural transmission. The article is methodologically rigorous and provides a genetic dimension to linguistic variation that is much needed in the field. As such it strikes me as a valuable contribution to the field.

I have two observations to make:

First, this finding depends on a previous article by the authors where they argue that variation in READ1 is significantly associated with number of consonants in languages. The authors have expanded the population panel from 43 to 51 and replicated this result. However, it's not totally transparent from the text how the present article builds upon previous results, including past limitations or knowledge gaps in this line of research. The article could benefit from a small statement justifying how the current work builds upon previous models by the authors.

Second, the logical link between DCDC2 variation and stop-consonant discrimination, crucial for providing a mechanistic explanation of the correlation, relies on experiments in mice. Although the authors are generally careful, the article would benefit from a few additional statements making clear that the reported correlations will eventually have to be demonstrated in humans.

Review form: Reviewer 2

Recommendation

Major revision is needed (please make suggestions in comments)

Scientific importance: Is the manuscript an original and important contribution to its field?

Excellent

General interest: Is the paper of sufficient general interest?

Excellent

Quality of the paper: Is the overall quality of the paper suitable?

Good

Is the length of the paper justified?

Yes

Should the paper be seen by a specialist statistical reviewer?

No

Do you have any concerns about statistical analyses in this paper? If so, please specify them explicitly in your report.

Yes

It is a condition of publication that authors make their supporting data, code and materials available - either as supplementary material or hosted in an external repository. Please rate, if applicable, the supporting data on the following criteria.

Is it accessible?

No

Is it clear?

Yes

Is it adequate?

Yes

Do you have any ethical concerns with this paper?

No

Comments to the Author

Dear authors and editor,

The paper was really pleasant to read, and the topic is very interesting. Thanks for this contribution !

Note that I am not able to judge the second half of the manuscript (from page 10 to 13), which is a contextualisation of the result in a field that I don't know. I suggest to let the paper be seen by a specialist of the DCDC2 knockout experiments, for instance.

I have several concerns about the statistical analysis (SI, page 4 : « Modeling procedure »). I would ask the authors to perform a better model selection, through the following steps. They then should verify that their results hold in this case, were they follow the best statistical practices.

- a) Instead of removing the data points that do not fit a model of normality of the residuals, a model of non-normality of the residuals should be used.
- b) A logit-like transformation of the data should be considered, knowing that the measures are bounded between 0 and 1. It could maybe allow to use normally distributed residuals, and ignore step a.
- c) The correction for language families was mandatory to control a biais between families. Nevertheless, highly correlated languages inside language families may also biais the analysis. I

would ask the authors to replicate their analysis, excluding the two families with lot of languages from the dataset (Indo-European and Afroasiatic).

d) An AIC should be use to compare the models.

Moreover, it is really important to give an open access to the data and the scripts used by the authors to perform the analyses. It is a good practice, in order to let the work be replicable for the community, and to allow the reviewers to check the methods and the results.

Overall, I highly recommend this article for its scientific importance, but the statistical flaws are too critical to be published without a major revision with a possibility of rejection.

Review form: Reviewer 3

Recommendation

Major revision is needed (please make suggestions in comments)

Scientific importance: Is the manuscript an original and important contribution to its field?

Good

General interest: Is the paper of sufficient general interest?

Good

Quality of the paper: Is the overall quality of the paper suitable?

Marginal

Is the length of the paper justified?

Yes

Should the paper be seen by a specialist statistical reviewer?

No

Do you have any concerns about statistical analyses in this paper? If so, please specify them explicitly in your report.

Yes

It is a condition of publication that authors make their supporting data, code and materials available - either as supplementary material or hosted in an external repository. Please rate, if applicable, the supporting data on the following criteria.

Is it accessible?

Yes

Is it clear?

Yes

Is it adequate?

Yes

Do you have any ethical concerns with this paper?

No

Comments to the Author

Please refer to comments from Editors

Decision letter (RSPB-2019-2712.R0)

03-Feb-2020

Dear Dr Tang:

Your manuscript has now been peer reviewed and the reviews have been assessed by an Associate Editor. The reviewers' comments (not including confidential comments to the Editor) and the comments from the Associate Editor are included at the end of this email for your reference. As you will see, the reviewers and the Editors have raised some concerns with your manuscript and we would like to invite you to revise your manuscript to address them.

Research ethics:

Use of animals and field studies:

It is a condition of publication that you make available the data and research materials supporting the results in the article. Datasets should be deposited in an appropriate publicly available repository and details of the associated accession number, link or DOI to the datasets must be included in the Data Accessibility section of the article

(<https://royalsociety.org/journals/ethics-policies/data-sharing-mining/>). Reference(s) to datasets should also be included in the reference list of the article with DOIs (where available).

Please submit a copy of your revised paper within three weeks. If we do not hear from you within this time your manuscript will be rejected. If you are unable to meet this deadline please let us know as soon as possible, as we may be able to grant a short extension.

Best wishes,

Dr Robert Barton

Associate Editor

Comments to Author:

Three reviewers have read your manuscript and express enthusiasm, but a number of major concerns and suggestions have been made, especially reviewers 2 and 3. Please revise the manuscript, as it will need to be reviewed again by these 2 reviewers.

Reviewer(s)' Comments to Author:

Referee: 1

Comments to the Author(s)

The article integrates data from Genetics and Linguistics to put forward an hypothesis about the effect of weak genetic biases in language diversity. In particular, the authors propose that variation in the READ1 regulatory element, in the DCDC2 gene, is correlated with stop-consonant discrimination. Small effects of genetics in auditory processing would then be amplified into language diversity by cultural transmission. The article is methodologically rigorous and provides a genetic dimension to linguistic variation that is much needed in the field. As such it strikes me as a valuable contribution to the field.

I have two observations to make:

First, this finding depends on a previous article by the authors where they argue that variation in READ1 is significantly associated with number of consonants in languages. The authors have expanded the population panel from 43 to 51 and replicated this result. However, it's not totally transparent from the text how the present article builds upon previous results, including past limitations or knowledge gaps in this line of research. The article could benefit from a small statement justifying how the current work builds upon previous models by the authors.

Second, the logical link between DCDC2 variation and stop-consonant discrimination, crucial for providing a mechanistic explanation of the correlation, relies on experiments in mice. Although the authors are generally careful, the article would benefit from a few additional statements making clear that the reported correlations will eventually have to be demonstrated in humans.

Referee: 2

Comments to the Author(s)

Dear authors and editor,

The paper was really pleasant to read, and the topic is very interesting. Thanks for this contribution !

Note that I am not able to judge the second half of the manuscript (from page 10 to 13), which is a contextualisation of the result in a field that I don't know. I suggest to let the paper be seen by a specialist of the DCDC2 knockout experiments, for instance.

I have several concerns about the statistical analysis (SI, page 4 : « Modeling procedure »). I would ask the authors to perform a better model selection, through the following steps. They then should verify that their results hold in this case, were they follow the best statistical practices.

- a) Instead of removing the data points that do not fit a model of normality of the residuals, a model of non-normality of the residuals should be used.
- b) A logit-like transformation of the data should be considered, knowing that the measures are bounded between 0 and 1. It could maybe allow to use normally distributed residuals, and ignore step a.
- c) The correction for language families was mandatory to control a bias between families. Nevertheless, highly correlated languages inside language families may also bias the analysis. I would ask the authors to replicate their analysis, excluding the two families with lot of languages from the dataset (Indo-European and Afroasiatic).

d) An AIC should be used to compare the models.

Moreover, it is really important to give an open access to the data and the scripts used by the authors to perform the analyses. It is a good practice, in order to let the work be replicable for the community, and to allow the reviewers to check the methods and the results.

Overall, I highly recommend this article for its scientific importance, but the statistical flaws are too critical to be published without a major revision with a possibility of rejection.

Referee: 3

Comments to the Author(s)

Several issues arise. The authors need to handle their three kinds of distance more openly and more subtly. They have genetic, linguistic and geographical distance. Their distance measures are crude. As an acceptable minimal step they should convert all of them to distance matrices using more gradations for distance than they have. Geographical could be Great Circle distances modified by likely water crossing points; linguistic should be more than just family level. It could even be based on a measure of sound differences; I think they have used RU1-1 allele frequencies for the genetic distance -- not sure. They should use their SNP distances either as a simple distance matrix or a matrix of F_{st} values.

Then, they should examine the relationship of each of these matrices to the matrices of sound differences, separately in Mantel tests. They should also examine Mantels among pairs of the three distances. This would give readers a concrete sense of what is going on, and whether all three matter.

Finally, and ideally, they should try to include these three distance matrices in their mixed model. This can be hard because I think software for including more than one distance matrix is limited, but they should make an effort. The results of the Mantels might suggest that one or more of the distance matrices is redundant or not important.

Aside: there is a temptation to ask them to build a genealogy from their genetic data (SNPs) and to use some sort of posterior set of these genealogies in a comparative methods framework to study the RU1-1 alleles. This could be insisted upon but would still have the problem of dealing with the other two distance matrices. The big issue here is whether the RU1-1 alleles have arisen independently. A genealogy could in principle tell us this. Lacking this approach, the authors need to be very careful in interpreting what they think any associations might mean. Currently they lean towards the genes driving language. But it could be the reverse and this should be made clear in their discussion.

Picky points: they don't tell us what the results were like including the Yemenite Jews, just that they deleted them following "model criticism". We need to know more here: if the whole story depends upon excluding this group, we need to know that.

The bootstrapping is good, but would be more persuasive if done with $N-k$ samples rather than $N-1$ samples. The latter tell us whether a single population matters, but we'd like to know if the results are even more robust. For example, so-called cross-validation is just a bootstrap

p where k corresponds to 50% of the samples being removed.

Latitude seems to be important but they don't explicitly control for that.

Finally, the style of writing is obtuse: sentences like that on lines 3-5 make me want to cry. And phrases like "model criticism" need to be reworked into simple English.

Author's Response to Decision Letter for (RSPB-2019-2712.R0)

See Appendix A.

RSPB-2019-2712.R1 (Revision)

Review form: Reviewer 1

Recommendation

Accept with minor revision (please list in comments)

Scientific importance: Is the manuscript an original and important contribution to its field?

Excellent

General interest: Is the paper of sufficient general interest?

Excellent

Quality of the paper: Is the overall quality of the paper suitable?

Good

Is the length of the paper justified?

Yes

Should the paper be seen by a specialist statistical reviewer?

No

Do you have any concerns about statistical analyses in this paper? If so, please specify them explicitly in your report.

No

It is a condition of publication that authors make their supporting data, code and materials available - either as supplementary material or hosted in an external repository. Please rate, if applicable, the supporting data on the following criteria.

Is it accessible?

Yes

Is it clear?

Yes

Is it adequate?

Yes

Do you have any ethical concerns with this paper?

No

Comments to the Author

Thanks for addressing my earlier comments into this new version of the manuscript

Review form: Reviewer 2

Recommendation

Accept as is

Scientific importance: Is the manuscript an original and important contribution to its field?

Good

General interest: Is the paper of sufficient general interest?

Excellent

Quality of the paper: Is the overall quality of the paper suitable?

Good

Is the length of the paper justified?

Yes

Should the paper be seen by a specialist statistical reviewer?

No

Do you have any concerns about statistical analyses in this paper? If so, please specify them explicitly in your report.

No

It is a condition of publication that authors make their supporting data, code and materials available - either as supplementary material or hosted in an external repository. Please rate, if applicable, the supporting data on the following criteria.

Is it accessible?

Yes

Is it clear?

Yes

Is it adequate?

Yes

Do you have any ethical concerns with this paper?

No

Comments to the Author

I thank the authors for the modifications, and congrat them for this really interesting paper.

Decision letter (RSPB-2019-2712.R1)

24-Apr-2020

Dear Dr Tang

I am pleased to inform you that your Review manuscript RSPB-2019-2712.R1 entitled "DCDC2 READ1 regulatory element: how temporal processing differences shape language" has been accepted for publication in Proceedings B. This is subject to only one change: one of the reviewers has requested that the title of your paper reflects the fact that one step in the chain of inference

from genes to language variation includes work on speech-sound discrimination in a rodent model. In light of this, the reviewer suggests using the modifier "may" (hence "how temporal processing differences may shape language"). This seems like a sensible suggestion to me.

The referee(s) do not recommend any other changes. Therefore, please proof-read your manuscript carefully and upload your final files for publication. Because the schedule for publication is very tight, it is a condition of publication that you submit the revised version of your manuscript within 7 days. If you do not think you will be able to meet this date please let me know immediately.

To upload your manuscript, log into <http://mc.manuscriptcentral.com/prsb> and enter your Author Centre, where you will find your manuscript title listed under "Manuscripts with Decisions." Under "Actions," click on "Create a Revision." Your manuscript number has been appended to denote a revision.

You will be unable to make your revisions on the originally submitted version of the manuscript. Instead, upload a new version through your Author Centre.

1) A text file of the manuscript (doc, txt, rtf or tex), including the references, tables (including captions) and figure captions. Please remove any tracked changes from the text before submission. PDF files are not an accepted format for the "Main Document".

2) A separate electronic file of each figure (tiff, EPS or print-quality PDF preferred). The format should be produced directly from original creation package, or original software format. Please note that PowerPoint files are not accepted.

3) Electronic supplementary material: this should be contained in a separate file from the main text and the file name should contain the author's name and journal name, e.g. `authorname_procb_ESM_figures.pdf`

All supplementary materials accompanying an accepted article will be treated as in their final form. They will be published alongside the paper on the journal website and posted on the online figshare repository. Files on figshare will be made available approximately one week before the accompanying article so that the supplementary material can be attributed a unique DOI. Please see: <https://royalsociety.org/journals/authors/author-guidelines/>

4) Data-Sharing and data citation

It is a condition of publication that data supporting your paper are made available. Data should be made available either in the electronic supplementary material or through an appropriate repository. Details of how to access data should be included in your paper. Please see <https://royalsociety.org/journals/ethics-policies/data-sharing-mining/> for more details.

<http://datadryad.org/submit?journalID=RSPB&manu=RSPB-2019-2712.R1> which will take you to your unique entry in the Dryad repository.

Once again, thank you for submitting your manuscript to Proceedings B and I look forward to receiving your final version. If you have any questions at all, please do not hesitate to get in touch.

Sincerely,
Dr Robert Barton
Editor, Proceedings B
<mailto:proceedingsb@royalsociety.org>

Reviewer(s)' Comments to Author:

Referee: 1

Comments to the Author(s)
Thanks for addressing my earlier comments into this new version of the manuscript

Referee: 2

Comments to the Author(s)
I thank the authors for the modifications, and congrat them for this really interesting paper.

Decision letter (RSPB-2019-2712.R2)

30-Apr-2020

Dear Dr Tang

I am pleased to inform you that your manuscript entitled "*DCDC2* READ1 regulatory element: how temporal processing differences may shape language" has been accepted for publication in Proceedings B.

Your article has been estimated as being 5 pages long. Our Production Office will be able to confirm the exact length at proof stage.

Open Access

You are invited to opt for Open Access, making your freely available to all as soon as it is ready for publication under a CCBY licence. Our article processing charge for Open Access is £1700. Corresponding authors from member institutions

Paper charges

Sincerely,
Proceedings B
<mailto:proceedingsb@royalsociety.org>

Appendix A

Dear Dr. Barton,

We wish to thank all three reviewers for their insightful comments. Below we provide detailed responses to all the comments and concerns, which have significantly improved the scientific rigor, transparency, and clarity of the manuscript. We also wish to thank you for the opportunity to revise. Please let us know if there are any further questions, which we would be happy to address.

Sincerely,
Kevin Tang, Ph.D.

Referee: 1

“First, this finding depends on a previous article by the authors where they argue that variation in READ1 is significantly associated with number of consonants in languages. The authors have expanded the population panel from 43 to 51 and replicated this result. However, it's not totally transparent from the text how the present article builds upon previous results, including past limitations or knowledge gaps in this line of research. The article could benefit from a small statement justifying how the current work builds upon previous models by the authors.”

>> Author's response:

We agree with the reviewer. To address these concerns, we have added the following to the introduction.

Page 3-4, lines 59-66:

“These studies suggest that through their effect on DCDC2 transcription, some READ1 alleles can enhance temporal precision and speech sound discrimination to favor retention or acquisition of selected consonantal contrasts. In the present study we build on and expand our previous results to investigate the relationship between the population frequency of RU1-1 and the numbers of specific manners of consonants used in a language. Consonant manners differ in their temporal and spectral characteristics. The perception of these different manners may be more or less sensitive to the differences in temporal processing conferred by RU1-1 specific regulation of DCDC2, favoring retention or acquisition of selected consonantal contrasts.”

“Second, the logical link between DCDC2 variation and stop-consonant discrimination, crucial for providing a mechanistic explanation of the correlation, relies on experiments in mice. Although the authors are generally careful, the article would benefit from a few additional statements making clear that the reported correlations will eventually have to be demonstrated in humans.”

>> Author's response:

We agree with the reviewer. We have added the following to clarify.

Page 11-12, lines 214-216:

“In addition, the link between DCDC2 expression and stop-consonant discrimination relies on published experiments in rodents. Further studies in humans are necessary to demonstrate a more direct genotype-phenotype connection.”

Referee: 2

“I have several concerns about the statistical analysis (SI, page 4 : « Modeling procedure »). I would ask the authors to perform a better model selection, through the following steps. They then should verify that their results hold in this case, were they follow the best statistical practices.

a) Instead of removing the data points that do not fit a model of normality of the residuals, a model of non-normality of the residuals should be used.

b) A logit-like transformation of the data should be considered, knowing that the measures are bounded between 0 and 1. It could maybe allow to use normally distributed residuals, and ignore step a.

c) The correction for language families was mandatory to control a biais between families. Nevertheless, highly correlated languages inside language families may also biais the analysis. I would ask the authors to replicate their analysis, excluding the two families with lot of languages from the dataset (Indo-European and Afroasiatic).

d) An AIC should be use to compare the models.”

>> Author's response

We thank the reviewer for pointing out the deficiencies in our statistical analyses. We previously misrepresented the normality of our residuals and we were overly conservative. The situation would be better characterized with a normal distribution, therefore there was no need to remove any data points. Inspired by these comments, as well as from Reviewer #3 below, we changed our approach (Methods and Results, page 5-9, lines 85-152, and SI Materials and Methods) to include tighter statistical controls. With the new model, the residuals are of a normal distribution (see the attached figure below, Figure 1), therefore it was no longer needed to remove data points or transform the data (points a and b above). As suggested in point d, we now use AIC for model comparisons (SI Modeling procedure). We achieved the same results and direction of effect, and nearly the same statistical significance (RU1-1/stops p-value = 0.0318; RU1-1/nasals p-value = 0.0798).

Regarding correction for bias within and across language families (point c above), we now correct for this confound due to linguistic relatedness by computing the degree of sound inventory overlap between populations, and incorporating the principal components into our regression model (Page 6, lines 103-105, Page 6-7, lines 117-120, and SI Linguistic relatedness, page 5, lines 87-97). Furthermore, inspired by point c and Reviewer #3's suggestion to ensure that the results were not driven primarily by a particular set of populations, we applied a leave- k -out method from $k = 1$ to 30 over the 51 populations. The effects remained the same for 95% of the subsets even when almost 50% ($k = 26$ for stops and $k = 23$ for nasals) were excluded, showing that the associations were not driven by particular subsets (Page 8, lines 138-142; SI page 7, lines 133-145, and SI Table 4).

Figure 1: A density plot of the residuals from the best regression model

“Moreover, it is really important to give an open access to the data and the scripts used by the authors to perform the analyses. It is a good practice, in order to let the work be replicable for the community, and to allow the reviewers to check the methods and the results.”

>> Author's response:

We agree and thank the reviewer for pointing out our oversight. The data and the scripts are now available on the Dryad Data Repository. The repository will be available to the public when the associated manuscript is accepted and the repository processed by the curators. The reviewers can access the repository via a temporary Dryad URL: https://datadryad.org/stash/share/yMtWZ82KevX4WRgt_ikwMOim9nGeRj0lVVzc_z3oKwkg. We amended the manuscript (Page 14, line 298-299) to include the public link as well as this temporary Dryad URL as a placeholder until the public link becomes available.

Referee: 3

“Several issues arise. The authors need to handle their three kinds of distance more openly and more subtly. They have genetic, linguistic and geographical distance. Their distance measures are crude. As an acceptable minimal step they should convert all of them to distance matrices using more gradations for distance than they have. Geographical could be Great Circle distances modified by likely water crossing points; linguistic should be more than just family level. It could even be based on a measure of sound differences; I think they have used RU1-1 allele frequencies for the genetic distance -- not sure. They should use their SNP distances either as a simple distance matrix or a matrix of Fst values.”

>> Author’s response:

We thank the reviewer for these suggestions which we have now incorporated into the analyses.

To correct for possible genetic confounding, as suggested we computed pairwise Fst values between populations (Page 6, lines 99-101; SI Page 3 Genetic Relatedness).

To correct for possible confounding due to geographic proximity, as suggested we now model migratory distances between populations and the putative location of human origin using great circle distances along with migratory waypoints (Page 6, lines 101-103; SI Page 4 Geographical proximity).

To correct for possible confounding due to linguistic relatedness, as suggested, we now use the sound inventory of the populations to compute the degree of inventory overlap between populations (Page 6, Page 103-105; SI Page 5 Linguistic relatedness).

“Then, they should examine the relationship of each of these matrices to the matrices of sound differences, separately in Mantel tests. They should also examine Mantels among pairs of the three distances. This would give readers a concrete sense of what is going on, and whether all three matter.”

>> Author’s response:

We agree with the reviewer. We have now examined the relationship of these matrices using Mantel tests. We have added the following to report the findings of the tests and to give readers a concrete sense of the relationship between the matrices.

Page 6, lines 106-112

“To better understand these three effects, we evaluated how genetic relatedness, geographical proximity, and linguistic relatedness correlate with each other using the Mantel test, a statistical test of the correlation between distance matrices. Statistical significance was determined using 1,000 permutations. Genetic relatedness and geographical proximity are the most strongly correlated with $r = 0.563$ ($p = 0.001$), while linguistic relatedness does not correlate with either genetic relatedness ($r = -0.0771$, $p = 0.860$) or geographical proximity ($r = 0.01434$, $p = 0.448$).”

“Finally, and ideally, they should try to include these three distance matrices in their mixed model. This can be hard because I think software for including more than one distance matrix is limited, but they should make an effort. The results of the Mantels might suggest that one or more of the distance matrices is redundant or not important.”

>> Author’s response:

Our mantel tests suggest that none of the three pairs are particularly correlated enough to be considered as redundant. We have added the following to report what the mantel tests suggest.

Page 6, lines 112-116

“The result of the Mantel tests suggests that all three effects should be taken into account in the modeling of the association between the five manners of articulation and RU1-1, given that linguistic relatedness is relatively independent from geographical proximity and genetic relatedness, and geographical proximity and genetic relatedness are only moderately correlated.”

“Aside: there is a temptation to ask them to build a genealogy from their genetic data (SNPs) and to use some sort of posterior set of these genealogies in a comparative methods framework to study the RU1-1 alleles. This could be insisted upon but would still have the problem of dealing with the other two distance matrices. The big issue here is whether the RU1-1 alleles have arisen independently. A genealogy could in principle tell us this.”

>> Author’s response:

We thank the reviewer for raising this question. In our previous paper, DeMille, et al, (2018), we estimated that the RU1 duplication event occurred sometime between 4 million to 700 thousand years ago, between the most recent common ancestor (MRCA) shared with chimpanzee and the MRCA shared with Neanderthal. The likelihood of RU1-1 occurring again in human history is extremely small as the rate for small deletions is about 5.3×10^{-10} to 5.8×10^{-10} per site per generation (Campbell, C. D., & Eichler, E. E. (2013). Properties and rates of germline mutations in humans. Trends in genetics: TIG, 29(10), 575-584. <https://doi.org/10.1016/j.tig.2013.04.005>). This means that the probability of an independent deletion at this site is estimated to be less than 0.01%. Our motivation

for dividing READ1 alleles into those with one copy of RU1, versus two copies of RU1 was that the odds that RU1-2 alleles changed to RU1-1 or vice-versa are negligibly small. Therefore, there would not be independent mutation events.

“Lacking this approach, the authors need to be very careful in interpreting what they think any associations might mean. Currently they lean towards the genes driving language. But it could be the reverse and this should be made clear in their discussion.”

>> Author’s response:

We think this is unlikely, but as the reviewer points out, it is possible. We therefore added the following to the discussion (Page 11-12, lines 216-222):

“It should also be noted that although we interpret the association between RU1-1 and stop-consonant description as a possible driver for phoneme inventory change among populations, a strict interpretation of the analysis means that the inverse relationship is also possible and that certain phoneme inventories may have favored genetic selection. However, we view this as unlikely, given arguments from computational simulations of language change and experimental evidence from human artificial language learning and song birds (27).”

“Picky points: they don't tell us what the results were like including the Yemenite Jews, just that they deleted them following "model criticism". We need to know more here: if the whole story depends upon excluding this group, we need to know that.”

>> Author’s response:

Since revising our modeling approach, we no longer drop out any populations from our analyses. In addition, thanks to the reviewer’s suggestion to perform bootstrapping with N-k samples (see below), the result remains the same with or without the Yemenite Jews (based on the bootstrapping result at $k = 1$).

“The bootstrapping is good, but would be more persuasive if done with N-k samples rather than N-1 samples. The latter tell us whether a single population matters, but we'd like to know if the results are even more robust. For example, so-called cross-validation is just a bootstrap where k corresponds to 50% of the samples being removed.”

>> Author’s response:

We thank the reviewer for the suggestions and have incorporated them into the revision (Page 8 lines 138-142; SI Page 7, lines 133-145; SI Table 4):

“To evaluate whether the associations were robust across different population subsets we performed a series of leave-k-out analyses by population from $k = 1$ (jackknife) to $k = 30$ (SI Tables S4). The directions of the beta-values remained

positive for stops and negative for nasals for 95% of the subsets even when almost 50% ($k = 26$ for stops and $k = 23$ for nasals) were excluded, showing that the associations were not driven by particular subsets.”

“Latitude seems to be important but they don't explicitly control for that.”

>> Author's response:

Latitude is incorporated into the correction for geographical proximity where we now model migratory distances using great circle distances along with migratory waypoints, which increases the accuracy of these distances (Page 6 lines 101-103; SI Page 4 Geographic proximity).

“Finally, the style of writing is obtuse: sentences like that on lines 3-5 make me want to cry. And phrases like "model criticism" need to be reworked into simple English.”

>> Author's response:

We thank the reviewer for making these suggestions to help us improve the readability of the manuscript. These have been deleted and text revised.